# Genomic Complexity and Complex Chromosomal Rearrangements in Genetic Diagnosis: Two Illustrative Cases on Chromosome 7

**DOI:** 10.3390/genes14091700

**Published:** 2023-08-27

**Authors:** Nicoletta Villa, Serena Redaelli, Stefania Farina, Donatella Conconi, Elena Maria Sala, Francesca Crosti, Silvana Mariani, Carla Maria Colombo, Leda Dalprà, Marialuisa Lavitrano, Angela Bentivegna, Gaia Roversi

**Affiliations:** 1UC Medical Genetics, Fondazione IRCCS San Gerardo dei Tintori, 20900 Monza, Italygaia.roversi@unimib.it (G.R.); 2School of Medicine and Surgery, University of Milan-Bicocca, 20900 Monza, Italy; 3Department of Obstetrics, Fondazione IRCCS San Gerardo dei Tintori, 20900 Monza, Italy; 4Neonatal Intensive Care Unit, Fondazione IRCCS San Gerardo dei Tintori, 20900 Monza, Italy

**Keywords:** complex chromosomal rearrangements, array-CGH, genomic complexity, chromosome 7, segmental duplications

## Abstract

Complex chromosomal rearrangements are rare events compatible with survival, consisting of an imbalance and/or position effect of one or more genes, that contribute to a range of clinical presentations. The investigation and diagnosis of these cases are often difficult. The interpretation of the pattern of pairing and segregation of these chromosomes during meiosis is important for the assessment of the risk and the type of imbalance in the offspring. Here, we investigated two unrelated pediatric carriers of complex rearrangements of chromosome 7. The first case was a 2-year-old girl with a severe phenotype. Conventional cytogenetics evidenced a duplication of part of the short arm of chromosome 7. By array-CGH analysis, we found a complex rearrangement with three discontinuous trisomy regions (7p22.1p21.3, 7p21.3, and 7p21.3p15.3). The second case was a newborn investigated for hypodevelopment and dimorphisms. The karyotype analysis promptly revealed a structurally altered chromosome 7. The array-CGH analysis identified an even more complex rearrangement consisting of a trisomic region at 7q11.23q22 and a tetrasomic region of 4.5 Mb spanning 7q21.3 to q22.1. The mother’s karyotype examination revealed a complex rearrangement of chromosome 7: the 7q11.23q22 region was inserted in the short arm at 7p15.3. Finally, array-CGH analysis showed a trisomic region that corresponds to the tetrasomic region of the son. Our work proved that the integration of several technical solutions is often required to appropriately analyze complex chromosomal rearrangements in order to understand their implications and offer appropriate genetic counseling.

## 1. Introduction

Complex chromosome rearrangements (CCRs) are currently defined as structural genome variations that involve more than two chromosome breaks and result in exchanges of chromosomal segments. CCRs have always been classified into different types based on their complexity and the number of chromosomes involved [1,2]. However, there is a large amount of confusion in the literature because there are many definitions of CCRs but none is adequate to include all types. Furthermore, thanks to the application of molecular techniques with increased resolution, increasing numbers of “simple” and/or balanced rearrangements, sometimes involving a single chromosome, such as inversions, distal deletions or duplications, and inverted duplications/deletions, are found to be unexpectedly “complex” and/or unbalanced [3,4]. Therefore, any formal standard definition of a CCR based on the number of chromosomes and the number of breaks is impossible. Accordingly, the proposal of Madan [5] “to describe a complex rearrangement as a CCR involving that and that chromosome, qualifying it as familial or *de novo*, balanced or unbalanced as appropriate”, could be the best way to describe a CCR [2].

The genetic and reproductive counseling of carriers for CCRs remains difficult despite advances in our understanding of the mechanisms involved in their formation. This path is fundamental because heterozygous carriers of CCRs have a high risk of spontaneous abortions and a child with multiple congenital abnormalities. Human oogenesis can deal with the complexity of CCRs and may generate phenotypically normal children. In contrast, the lack of transmitting males is mostly caused by spermatogenesis impairment or arrest, which is commonly linked to CCRs and results in sterility or subfertility [1,6].

One of the best-known mechanisms to cause CCRs is non allelic homologous recombination (NAHR), mediated by the presence of region-specific low copy repeats (LCR) [7,8]. However, in the past decade, breakpoint sequencing of some CCRs led to the discovery of a novel “one-step” chaotic chromosomal rearrangement described by Liu et al. [9] as chromoanasynthesis. This latter is based on DNA replication defects and involves multiple template-switching events driven by microhomology-mediated break-induced replication (MMBIR) or fork stalling and template switching (FoSTeS) mechanisms, with the formation of highly rearranged chromosomes. With respect to the other two types of chromoanagenesis, i.e., chromothripsis and chromoplexy, which represent very chaotic mutational processes in cancer evolution, chromoanasynthesis may occur in germlines or during early embryonic development and can lead to the formation of stable and heritable rearranged genomic constitutions [10].

In this work, we describe two unrelated pediatric patients with complex rearrangements and an abnormal phenotype. With reference to the proposal of Madan [5], both involve chromosome 7 and are unbalanced. The first case was a *de novo,* apparently simple, partial duplication of the 7p arm identified by conventional cytogenetics that, after array-CGH analysis, was redefined as being complex with three distinct duplications. The second case was a recombinant chromosome 7 of an already aberrant maternal chromosome, solved with the help of array-CGH. In both cases, we suggested the presumed rearrangement mechanism thanks to in silico analysis of the segments involved in the breakpoints. Both cases demonstrate that the combination of different techniques is decisive for achieving a correct diagnosis of a CCR, so we consider them instructive cases.

## 2. Materials and Methods

### 2.1. Chromosome Analysis

Peripheral blood metaphases were obtained from phytohaemagglutinin-stimulated lymphocytes, cultured with Synchro kit (Celbio Euroclone S.p.A., Pero, Milano, Italy) according to the manufacturer’s protocol and as previously reported [3]. Chromosome analysis was carried out by applying QFQ and GTG banding according to routine procedures, and all the karyotypes were reconstructed following the guidelines of ISCN 2020 [11].

### 2.2. FISH Analysis

Fluorescence in situ hybridization (FISH) was carried out according to the manufacturer’s protocol. The following probes were applied: Williams region (*ELN*, 7q11.23) with the control probe in 7q31 (AbbottVysis); *EGFR* (7p11.2) with the centromeric probe D7Z1; *CUTL1* (7q22) with the control probe in 7q36 (Kreatech); Multiprobe Octochrome Kit (Cytocell, Cambridge, UK) containing whole painting probes for all chromosomes; and 7p and 7q subtelomere probes (Oncor, Gaithersburg, MD, USA). Probes were labeled in different colors with three spectrally independent fluorophores (SpectrumGreen, SpectrumOrange, and Spectrum Aqua). Moreover, eight different BAC (Bacterial Artificial Chromosome) probes were co-hybridized with the Williams region probe. BAC probes were generated by nick translation and labeled with biotin-dUTP (Deoxyuridine triphosphate) (Roche, Basel, Switzerland). All digital images were captured using a Leitz microscope (Leica DM5000B, Leica Microsystems GmbH, Leica Microsystems, Milan, Italy) equipped with a charge coupled device (CCD) camera (Hamamatsu) and analyzed using software Chromowin-Plus (Cromowin Tesi Imaging). All images were captured at 1000× magnification.

### 2.3. Array Comparative Genomic Hybridization (Array-CGH)

Genomic copy number analysis was performed with array-CGH using the Human Genome CGH Microarray kit 105A (Agilent Technologies) following the manufacturer’s recommendations. The target DNA was extracted from peripheral blood using the Wizard Genomic DNA purification kit (PromegaTM, Mannheim, Germany). The DNA control reference was a sex-matched DNA pool recommended by Agilent and provided by Promega (p/n G1521, female; p/n G1471, male). The arrays were scanned at 2 μm resolution using an Agilent microarray scanner (G2565CA) and analyzed using CyoGenomics 3.0 software (Agilent Technologies, Palo Alto, CA, USA). The aberration detection method 2 (ADM-2) algorithm was used to compute and assist in the identification of aberrations for a given sample. Significant chromosomal aberrations were determined using the algorithm ADM-2 with a threshold of 5 and a minimum absolute average log2 ratio of 0.25. Putative chromosome copy number changes were defined by intervals of 3 or more adjacent probes and were considered as being duplicated or deleted when results exceeded |0.25|. All nucleotide positions were based on the Human Reference Sequence Assembly, February 2009 GRCh37/hg19 of the UCSC Genome Browser (http://genome.ucsc.edu/, last access date: 11 November 2022). All the molecular karyotypes were reconstructed following the guidelines of ISCN 2020 [11].

### 2.4. In Silico Analysis

Copy number alteration (CNA) regions were analyzed using in silico databases: Database of Genomic Variants (http://dgv.tcag.ca/dgv/app/home, last access date: 11 November 2022); Database of Chromosomal Imbalance and Phenotype in Humans using Ensemble Resources (DECIPHER, https://www.deciphergenomics.org/, last access date: 11 November 2022); and Clinical Variants NCBI (https://www.ncbi.nlm.nih.gov/clinvar, last access date: 11 November 2022). UCSC genome browser (https://genome.ucsc.edu/, last access date: 11 November 2022) was used in order to map the CNAs and to search for segmental duplications and repeating elements. Breakpoints regions were analyzed with BLAST (https://blast.ncbi.nlm.nih.gov/Blast.cgi, last access date: 11 November 2022).

## 3. Results

### 3.1. Case 1

#### 3.1.1. Clinical Presentation

A baby girl was born to healthy and unrelated parents after an uncomplicated pregnancy. Birthweight was 3.190 kg (50th p.), length was 50 cm (50–85th p.), and occipitofrontal circumference (OFC) was 34 cm (50th p.); the neonate’s Apgar scores were 9 and 10 at 1 and 5 min, respectively. A summary of clinical features is presented in Table 1. At clinical evaluation, the baby showed bifid uvula, cleft palate, and right choanal stenosis. From the first days of life, she was symptomatic of gastroesophageal reflux. At two years old, the weight was 11 kg (15–50th p.), the height was 86 cm (50th p.), and the OCF was 47 cm (50th p.). Physical examination revealed dysmorphic notes in the limbs and face with low hairline on the forehead, hypotelorism, small mouth, tendency to turricephaly, acromicria, clinodactyly of the 5th finger and the 2nd right toe, and bilateral syndactyly of the 2nd–3rd toe. Moreover, she presented hypotonia, ligamentous hyperlaxity, and ataxia. She had global delayed neurological development: she was able to maintain a sitting position and crawl by the age of 9 and 20 months, respectively; she walked at 24–26 months and spoke her first words at 15–18 months of age. She demonstrated mild intellectual disability with significant speech impairment. She showed psychomotor instability and attention deficit; however, the brain MRI, the ECG, and the echocardiography were normal. At the age of 8 she was diagnosed with a behavioral disorder characterized by hyperactivity, aggression towards herself and others, and stereotyped mannerisms. She was minimally verbal with language difficulties, supplemented by non-verbal communication, while hand-eye coordination and fine manual dexterity were acquired and preserved. She developed generalized, drug-resistant seizures.

#### 3.1.2. Laboratory Investigation

As a first step of analysis, we performed a conventional karyotype, which highlighted a possible duplication on the short arm of chromosome 7 (Figure 1A). Then, via FISH analysis, we confirmed that the rearrangement involved regions belonging exclusively to chromosome 7 and was intrachromosomal, without apparent loss of the terminal region (Figure 1A). In addition, we better defined the region of duplication at 7p21.3 and 7p21.2 bands using FISH analysis with BAC-specific probes (Figure 1B).

The karyotype was 46,XX,dup(7)(p?21p?15).ish dup(7)(wcp7+,p22+).

Finally, the array-CGH analysis highlighted a complex rearrangement consisting of three interstitial discontinuous duplicated regions affecting the short arm of chromosome 7 (Figure 1C). The molecular karyotype was as follows:

arr[GRCh37]7p22.1p21.3(7092947_8122771)x3,7p21.3(8791957_9757762)x3,7p21.3p15.3(11521133_22357359)x3.

Moreover, in silico analysis showed a complex situation full of repeating elements, especially short interspersed elements (SINEs) and segmental duplications (SDs), particularly concentrated near the BP1 and BP6 (Appendix A).

### 3.2. Case 2

#### 3.2.1. Clinical Presentation

A late preterm male infant was born by vaginal delivery at a gestational age of 36 weeks. Pregnancy was complicated by bilateral hydroureteronephrosis at the 20th gestation week, mildly dilated gut loops at the 28th gestation week, and poor fetal growth by the 32nd gestation week. A summary of clinical features is presented in Table 1. The child had a healthy 4-year-old sister. No family history of genetic disorders was reported. His parents were both healthy, but his father underwent infertility treatments, and his mother experienced a delay in age-appropriate milestones and speech without a specific diagnosis; moreover, they reported a previous pregnancy with a miscarriage at 7 gestational weeks. His birthweight was 2420 gr (<3rd p.), his length was 48 cm (15th p.), and his occipitofrontal circumference (OFC) was 32 cm (3rd p.); the neonate’s Apgar scores were 7 and 8 at 1 and 5 min, respectively. Physical examination revealed hypotonia, micrognathia, low ear implantation, left retroauricular peduncle, and bilateral cryptorchidism; abdominal ultrasound evaluation showed bilateral renal dysplasia. At 8 months, the baby’s weight was 7850 gr (15th p.), his length was 70 cm (50–15–50th p.), and the OFC was 46 cm (85th p.). Furthermore, he initiated social interactions and speech development with lallation was present. He was able to maintain a seated position with support; cryptorchidism completely regressed, while renal dysplasia was unchanged.

#### 3.2.2. Laboratory Investigation

The cytogenetic analysis evidenced a structural aberration of the chromosome 7p arm. The paternal karyotype was normal, while the maternal one showed an anomalous chromosome 7, but was cytogenetically distinguishable from that of the son (Figure 2). On first examination, the maternal one seemed to be a pericentric inversion of the region between the 7p15.3 and 7q22 bands. Via FISH analysis with locus-specific probes, we first excluded rearrangements with other chromosomes (Figure 2B); moreover, the nearly overlapping signals of the ELN (7q11.23) and control (7q31) probes on the anomalous chromosome 7 suggested the excision of some intervening sequences (Figure 2C). Subsequently, we proved that the rearrangement was an insertion and not a pericentric inversion because the red EGFR probe signal (7p11.2) had not moved from its correct position on the short arm of the abnormal chromosome (Figure 2A). Therefore, the region between 7q11.23 and 7q22 bands had been translocated and inserted into the short arm, at band 7p15.3, after undergoing a 180° rotation, consistent with signals observed in FISH (Figure 2A,C). Therefore, the maternal karyotype was as follows:

46.,XX,ins(7)(p15.3q22q11.23).ish (wcp7+,EGFR+,ELN+).

The abnormal chromosome 7 of the child is the result of a further rearrangement that occurred during maternal meiosis. In fact, the ELN (7q11.23) and control probe (7q31) signals appeared in the correct position on both chromosomes 7 of the child (Figure 2D), while the CUTL1 probe (7q22) revealed an anomalous additive signal on the terminal 7p of the abnormal chromosome (Figure 2E). Thus, the karyotype of the child was as follows:

46.,XY,rec(7)dup(7q)ins(7)(p15.3q22q11.23)dmat.ish rec(7)dup(7q)(ELN+,CUTL1++).

The array-CGH analysis revealed an imbalance of genetic material. In the mother, a trisomic region of 4.47 Mb embedded in the 7q22 band was unexpectedly identified. Her molecular karyotype was as follows:

arr[GRCh37] 7q21.3q22.1(97909997_102428256)x3.

In the child, the trisomic region of 4.47 Mb appeared as a tetrasomy; moreover, the trisomic region 7q11.23q22, evidenced by the karyotype, was confirmed (Figure 3). His molecular karyotype was as follows:

arr[GRCh37]7q11.23q21.3(76924142_97852766)x3,7q21.3q22.1(97909997_102428256)x4,7q22.1q22.3(102448348_106403919)x3.

Then, the array-CGH analysis of the maternal sample revealed an imbalance in 7q22.1 of 4.47 Mb, even if it was unable to specify its location, i.e., on the site of origin (long arm) or destination (short arm) of the insertion (Appendix A). However, since the child inherited the same copy number alteration but in a tetrasomy state (Figure 3), we can conclude that its correct location is on the short arm. Thus, the child inherited one normal chromosome 7 from his father and one recombinant 7 from his mother, consisting of the aberrant short arm (with the insertion) and the normal maternal q arm. This consideration assumes a recombination event between the two homologous 7s during maternal meiosis. In this case, the in silico analysis also showed a particular genomic complexity. In particular, we found a 6 Kb sequence in 7p15.3 with more than 97% homology with a sequence in 7q22, near the telomeric breakpoint of the copy number alteration of 4.47 Mb. In addition, the UCSC genome browser showed 98–99% similarity between segmental duplications at 7q11.23 and 7q22 breakpoints (Appendix A).

## 4. Discussion

Complex chromosomal rearrangements are rare, but their identification and study may be important as the associated phenotypic spectrum is highly variable and ranges from normal to infertility and/or intellectual disability and/or congenital abnormalities [1,2,3,4,5,6]. The application of advanced molecular techniques for the characterization of CCRs revealed that the rearrangements could be more complex than initially assumed by conventional cytogenetics [4] and could reveal unexpected imbalances, as in our cases. Segmental duplications (SDs) or duplicons or low copy repeats (LCRs) represent paralogous segments usually greater than 10 kb in length and with over 97% sequence identity [8]. These sequences have been shown to mediate recurrent rearrangements by non-allelic homologous recombination (NAHR) and stimulate nonrecurrent rearrangements [12,13].

Chromosome 7 contains a particular quantity of SDs (8.2%, 12.588 Kb), with marked differences between the two arms [14]. In particular, 7% of the sequences on chromosome 7 are intra-chromosomal duplications, while just 2% of the sequences are inter-chromosomal duplications, grouped mainly within the pericentromeric and subtelomeric regions of the short arm [15,16].

Here, we describe two cases of complex rearrangements with a very likely involvement of SD. Although we have not delved further into molecular investigations of breakpoints in our cases, we can discuss potential mechanisms based on data interpretation, database consultation, and in silico analyses to interpret them.

The three interstitial non-contiguous duplicated regions in the short arm of chromosome 7 of case 1 could be explained by non-allelic homologous recombination (NAHR) events that occur between two directly oriented paralogous SDs of the two homologous during meiosis in one of the two parents [17]. We can speculate that one duplication, for example, the largest one, in 7p21.3-15.3, occurred during prophase I in one of the two gametogenesis processes. Then, because the region is rich in repeated sequences (i.e., SINEs and LINEs) that support genome instability, subsequent postzygotic events may have created additional duplications by repairing breaks that emerged on the aberrant chromosome containing the large duplication deletions (Appendix A). The final aberrant chromosome could be the result of a positive selection over other less vital events. However, the existence of the configuration duplication–normal–duplication (dup-nml-dup) copy number states, typical of chromoanasynthesis, would suggest an alternate interpretation involving a “one-step” chaotic chromosomal rearrangement [9]. This mechanism is characterized by segments of extended microhomology and templated insertions, which are produced by errors in DNA replication, most notably fork stalling and template switching (FoSTeS) [18] and microhomology-mediated break-induced replication (MMBIR) [19] detectable only by sequencing the breakpoints. The patient’s phenotype was in agreement with the 7p duplication syndrome, or partial trisomy 7p, with hypotonia, skull anomalies, intellectual disability, and severe psychomotor delay in the acquisition of motor milestones. The phenotypic spectrum of dup 7p syndrome varies among patients according to the size of the duplicated segment, but determination of a critical region has been attempted. Alfardan et al. [20], Stankiewicz et al. [21], and Mégarbané et al. [22] reported similar cases of partial 7p trisomy with a phenotype comparable to our case 1. All authors discussed the possible role of *TWIST1 locus* at 7p21.2 in abnormal skull development, including the large anterior fontanel. The *TWIST* gene encodes a transcription factor of the basic helix–loop–helix protein family and plays an important role in mesodermal cell determination. In particular, it is involved in membranous ossification occurring during frontal, parietal, and malar bone formation. In humans, haploinsufficiency of *TWIST1* has been associated with Saethre–Chotzen syndrome, which is characterized by, among other clinical signs, craniosynostosis, the opposite of the delayed closure of fontanels. Therefore, it has been suggested that trisomy 7p cases with delayed closure of fontanels can be a result of the *TWIST* gene dosage effect. In addition, in a recent study, Romanelli Tavares et al. [23], studying auriculocondylar polygenic syndrome (ARCND), identified a unique 430 Kb tandem duplication at the *HDAC9/TWIST1 locus* that may cause deregulation of *TWIST1* expression, interfering with its regulatory elements. By impairing osteogenic differentiation and neural crest migration, this copy number variant led to the development of ARCND characteristics. Regarding *HDAC9*, it has not been associated with neural crest proliferation/migration or the specification of craniofacial elements. One last interesting gene included in the large duplication at 7p21.3 of case 1, potentially causative of the phenotype, and that deserves to be mentioned, is the transmembrane protein 106 B (*TMEM106B*). *TMEM106B* localizes to late endosomes and lysosomes, and is involved in dendrite morphogenesis and maintenance by regulating lysosomal trafficking via its interaction with MAP6 [24]. Its overexpression is associated with familial frontotemporal lobar degeneration [25].

Considering case 2, the high homology evidenced by BLAST analysis in the regions where the breakpoints fall in 7p15.3 and 7q22 would suggest that the insertion in the maternal chromosome occurred through a probable NAHR event during the meiosis of one of her parents. In addition, the UCSC genome browser showed 98–99% similarity between segmental duplications at 7q11.23 and 7q22 breakpoints (Appendix A). This complex genome architecture may have triggered a further event of NAHR at 7q11.23-q22.3 during maternal meiosis. In addition, the child’s recombinant chromosome 7 can be explained through a mechanism that predicts the formation of a loop in both homologues during a synapse. In the present case, we can assume an incomplete pairing of the two maternal chromosomes 7, with a loop on both homologues corresponding to the region 7q11.23q22. In the case of a single crossing-over in the centromeric region (Figure 4), two possible recombinants are created, one deleted and the other duplicated, as in case 2. Many cases are reported in the literature with duplication or partial trisomy of 7q, identified by chromosomal banding [26,27,28]. Many of these cases are derivatives of translocations and therefore have simultaneous deletions of the partner chromosome, as well as cases in which partial duplication/trisomy is combined with deletions of chromosome 7 [29,30]. Therefore, the comparison of the clinical phenotype is almost impossible due to the differences in the copy number alterations described from time to time. However, a literature review of CCR cases highlighted two studies exclusively involving chromosome 7 [31,32]. A patient with Silver–Russell syndrome (SRS) features who carried a duplication in 7p13-p11.2 in addition to a deletion in 7q11.21-q11.23 was identified by Catusi et al. [31]; while a *de novo* duplication of 7p21.1p22.2 in a child with autism spectrum disorder and craniofacial dysmorphism was described by Udayakumar et al. [32]. Our case 1 has neither deletions nor any of the symptoms characteristic of SRS, nor is there a significant overlap with the case of autism.

Starting from the description of a pure dup7q case observed in cytogenetics and further analyzed using array-CGH, Alfonsi et al. [26] reviewed the division into classes proposed 30 years earlier. The authors suggested that the comparison between clinical pictures could only be undertaken on a cytogenomic basis. In addition, the DECIPHER database reported some cases that overlap with Alfonsi’s case and our case 2 (Appendix A), but their clinical comparison appears to be useless because most of the clinical signs are non-specific and are shared with many other chromosomal pathologies. The interesting aspect of the case is the copy number variation in the 7q21.3q22 region also found in the mother, who, however, presented some clinical signs.

## 5. Conclusions

In conclusion, chromosome 7, which is abundant in segmental duplications and repetitive sequences, is prone to simple and complex structural rearrangements and can thus be defined as a fragile chromosome. However, from a clinical perspective, it will be necessary to better characterize the roles of the genes involved to correlate them with carriers’ phenotypes. Finally, we reported two illustrative cases of CCRs on chromosome 7 investigated using multiple techniques in order to understand their implications and offer appropriate genetic counseling.

## Figures and Tables

**Figure 1 genes-14-01700-f001:**
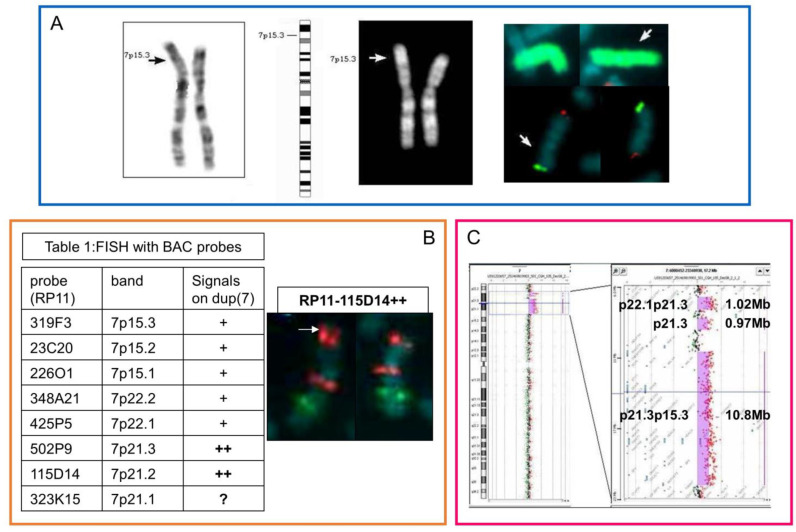
Case 1. (**A**) (from left to right): GTG and QFQ banded homologous chromosomes 7 of the proband, with the chromosome 7 ideogram in the middle; the arrows indicate the point of insertion. In the square on the right: FISH with whole chromosome 7 painting (top) and subtelomeric-specific probes of chromosome 7 (bottom, green 7ptel and red 7qtel). (**B**) BAC probes and their mapping region (band) and FISH results on chromosome dup(7) (+ single signal, ++ double signal) (Table 1); in the square on the right: FISH with RP11-115D14 (7p21.2, red distal signal), and the ELN probe (7q11.23, red signal) with the 7q22 control probe (green signal). The arrow indicates chromosome 7 with a double BAC probe signal. (**C**) Array-CGH profile, with an enlargement of the duplicated region.

**Figure 2 genes-14-01700-f002:**
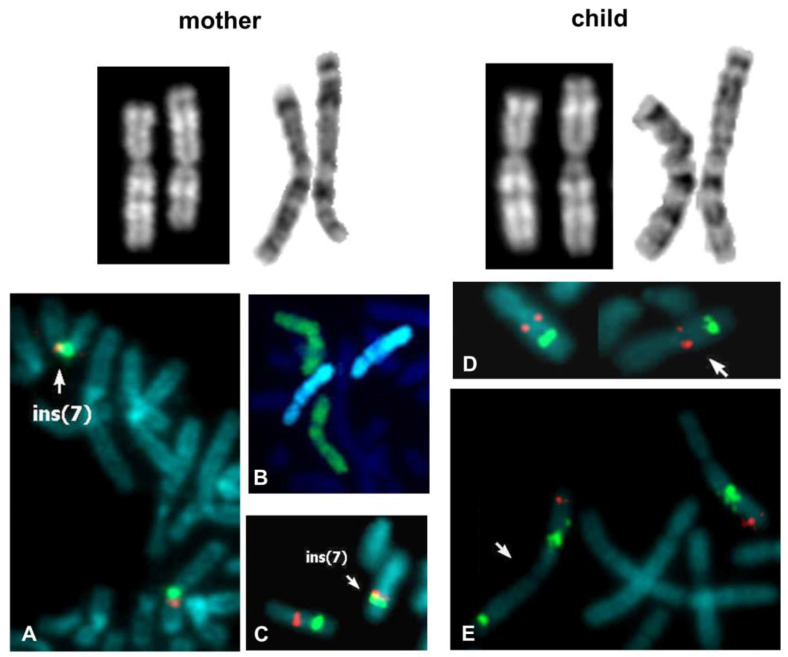
Case 2. Left: QFQ and GTG banded chromosomes 7 of the mother; the ins(7) chromosome is on the right. (**A**) FISH with D7Z1 (green signal) and *EGFR* (7p11.2, red signal) probes; (**B**) FISH with wcp7 (green chromosomes); (**C**) FISH with *ELN* (7q11.23, red signal) and the 7q22 control probes (green signal). The signal proximity on the ins(7) chromosome shows that the 7q21.3q22.2 region is excised. Arrows indicate ins(7) chromosomes. Right: QFQ and GTG banded chromosomes 7 of the child; the rec(7)dup(7q) chromosome is on the right. (**D**) FISH with *ELN* (7q11.23, red signal) and the 7q22 control probes (green signal) show a normal hybridization pattern; (**E**) FISH with *CUTL1* (7q22, green signal) and 7q36 (red signal) probes shows an anomalous additive signal for the *CUTL1* probe on the p arm of the rec(7) (arrowed).

**Figure 3 genes-14-01700-f003:**
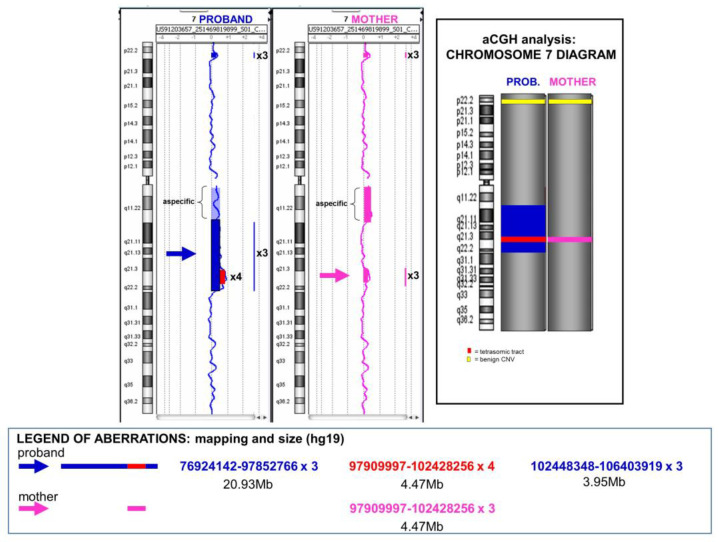
Case 2. Array-CGH profiles of the proband (on the left) and of the mother (on the right). Trisomic regions (in blue or pink) and tetrasomic regions (in red) are indicated. The ideogram of chromosome 7 and the respective alterations in the child’s and the mother’s chromosomes are schematized in the box on the right. A benign copy number variant (CNV) was also reported at 7p22.2 in both samples (in yellow). Bottom: reciprocal mapping and dimensions of the alterations identified by array-CGH analysis.

**Figure 4 genes-14-01700-f004:**
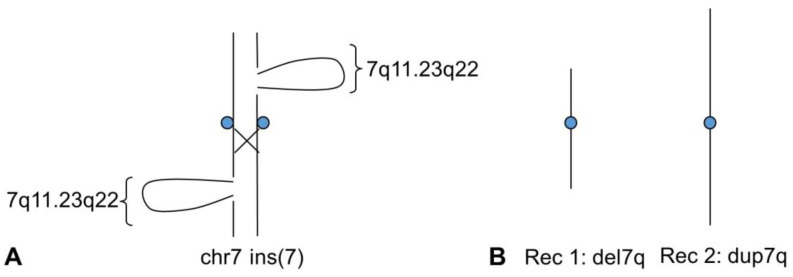
Diagram showing the suggested mechanism of formation of the recombinant chromosome of the child in case 2. (**A**) the normal (left) and the ins(7) (right) in maternal prophase 1. An incomplete pairing is supposed to be due to the region inserted at 7q11.23q22, which causes a loop in both chromosomes. (**B**) Meiotic recombinant chromosomes 7 following a single crossing-over event in the region near the centromere, Rec1 with 7q11.23q22 deletion and Rec2 with 7q11.23q22 duplication found in the child. The figure shows only one chromatid for each chromosome.

**Table 1 genes-14-01700-t001:** Summary of clinical features by category for the two patients.

Case	Case 1	Case 2
Age at diagnosis (Days, Months, Years)	28 m	5 d
Sex (M/F)	F	M
**Major Malformation**		
Criptorchid testes		**+**
Heart defect	−	
Renal abnormalities	−	**+**
Cerebral Abnormalities (MRI)	−	
**Minor Anomalies**		
Prominent/high forehead	−	
Hypertelorism	−	
Broad and/or flat nasal bridge		**+**
High palate	+	
Thin lip	+	
Micrognathia	+	
Abnormal palmar creases	+	
**Skeletal anomalies**		
Short neck		**+**
Digital anomalies	+	
**Medical Complications**		
Hypotonia	+	
Ataxia	+	
Epilepsy	+	
**Growth and Development**		
Small at birth	−	
Developmental delay	+	
Stereotypic behavior	+	
Mental retardation	+	
Language impairment	+	

(+): present sign; (−): absent sign; empty cells indicate the absence of specific information.

## Data Availability

The datasets generated during and/or analyzed during the current study are available from the corresponding author on reasonable request.

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
