# Peer review of "Genomic Complexity and Complex Chromosomal Rearrangements in Genetic Diagnosis: Two Illustrative Cases on Chromosome 7"

_genes, 2023, doi:10.3390/genes14091700_

Round 1

Reviewer 1 Report

The manuscript by Villa et al., entitled: 'Genomic complexity and complex chromosomal rearrangements in genetic diagnosis: example from two emblematic cases on chromosome 7' concerns an interesting description of two novel cases of CCR with chromosome 7 that can be a valuable data for genetic counseling.

However, there are some pints that should be improved by the authors before further steps of processing:

'emblematic' in the title sound strange and funny - maybe it should be changed for something more appropriate

22: position of what? gene?

25, 28, and further through the text: avoid 'anomaly, abnormality, situation' - it should be word 'rearrangement' each time (comes from CCR definition)

26: phenotype of what?

27-34: make some logic order of description (do not mix as now):  first - baby, second - mother

35: what 'strategies'?

42-45: not clear sentence

Introduction: lacks short description and definition of types of CCRs (I-IV), followed by appropriate references (i.e. Madan, 2012; Pellestor 2011; Poot, 2015); the same for statistics of infertile patients (line 54; i.e. Olszewska, 2020 for males or Madan, 2012 for both genders)

latin word (locus, etc.) and gene names should be written with coursive

69: to what type of CCR belonged the described cases?

77, 82, 99: brief protocols description needed

88-90: what kit has been used for preparation of BACs with conjugates?

94: '100x magnification': figures rather show 1000x magnification; maybe 100x objective? please, correct that

means of software: what exactly do you mean?

101: add cat. no. for Promega DNA

102: what model of the scanner?

151, 197/198: please, show the whole view of all chromosomes to be allowed for a statement that it was only from chr7 (can be as supplementary data)

161: 'on the other hand' suggests something opposite, not just additional data (remove or rephrase)

214: not 'situation' but rather 'amount of genetic material'

216, 220: whose?

254: add 'suggested' between 'the' and 'mechanism'

263: add at least appropriate 5 references

338, 340, 343: add more references

356-357: should be rewritten because it suggests the development of novel method (authors have not develop any new step-by-step procedure or technique)

371: number of ethical agreement has to be listed here

Please, additionally prepare review table of CCRs with chromosome 7 with all literature data available - then it will be very informative for the genetic counseling and suitable for 'Genes'. Of course, prepare short discussion of data reviewed according to your cases described in the manuscript.

English language sound quite fine but should be corrected by professional native speaker with biological background to avoid i.e. 'emblematic' as in the title and to get some better flow (grammar is some parts needs attention)

Reviewer 2 Report

The Authors reported on 2 cases of patients carrying complex abnormalities of chromosome 7,
having a severe clinical presentation. Complex chromosomal rearrangements are rarely compatible with survival, and therefore these 2 cases are highly informative with respect to both clinical characteristics and genetic complex rearrangement features. Hence, the manuscript is of interest.

I would first suggest revising the abstract to optimize spelling/grammar and quality of written English.

Few comments:

Abstract

1)     Line 24: mechanism’s identification. Please correct.

2)     Line 25: possible recurrence. Recurrence? Of what?

3)     Line 26: clinical picture. Please rephrase.

4)     Line 35: both strategies. Please specify which strategies.

5)     Maybe point out the clinical cases presented are paediatric cases.

Introduction

1)     Why did you choose to report on chromosome 7 CCRs. Any specific reason? Any particular characteristic that makes it of interest for the readers? Please explain.

I would first suggest revising the abstract to optimize spelling/grammar and quality of written English.

Reviewer 3 Report

My suggestions:

1. Were there MRIs or any brain imaging performed on the affected children?

2. Were the children, the authors present related or unrelated? Authors should clarify this. 

3. I would add a table, which summarizes the phenotypes of the two children, especially if they carried the same type of chromosomal rearrangements. 

4. Are there any environmental risk factors for chromosome 7 rearrangements? Such as smoking, pollution, poor diet, etc. Authors may discuss it. 

5. Is there any CSF biomarker for chromosome 7 rearrangements? Authors may discuss it briefly, if yes. 

Round 2

Reviewer 3 Report

Thank you, the authors fulfilled my suggestions.